# Aflatoxin Exposure in Immunocompromised Patients: Current State and Future Perspectives

**DOI:** 10.3390/toxins17080414

**Published:** 2025-08-16

**Authors:** Temitope R. Fagbohun, Queenta Ngum Nji, Viola O. Okechukwu, Oluwasola A. Adelusi, Lungani A. Nyathi, Patience Awong, Patrick B. Njobeh

**Affiliations:** 1Department of Biotechnology and Food Technology, University of Johannesburg, Doornfontein Campus, Johannesburg 2094, South Africa; njiqueenta86@gmail.com (Q.N.N.); solaceonline2009@gmail.com (O.A.A.); 2Department of Environmental, Water and Earth Sciences, Tshwane University of Technology, Arcadia Campus, Private Bag X680, Pretoria 0001, South Africa; viola.okechukwu15@gmail.com; 3Centre for Innovative Food Research (CIFR), Department of Biotechnology and Food Technology, University of Johannesburg, Doornfontein Campus, Johannesburg 2094, South Africa; lunganianyathi@gmail.com; 4Department of Health Sciences, University of Johannesburg, Doornfontein Campus, Johannesburg 2094, South Africa; patiencemaghah85@gmail.com

**Keywords:** aflatoxins, immunocompromised patients, exposure route, aflatoxin biomarkers, detection methods

## Abstract

Aflatoxins (AFs), harmful secondary metabolites produced by the genus *Aspergillus*, particularly *Aspergillus flavus* and *Aspergillus parasiticus*, are one of the best-known potent mycotoxins, posing a significant risk to public health. The primary type, especially aflatoxin B_1_ (AFB_1_), is a potent carcinogen associated with liver cancer, immunosuppression, and other health problems. Environmental factors such as high temperatures, humidity, and inadequate storage conditions promote the formation of aflatoxin in staple foods such as maize, peanuts, and rice. Immunocompromised individuals, including those with HIV/AIDS, hepatitis, cancer, or diabetes, are at increased risk due to their reduced detoxification capacity and weakened immune defenses. Chronic exposure to AF in these populations exacerbates liver damage, infection rates, and disease progression, particularly in developing countries and moderate-income populations where food safety regulations are inadequate and reliance on contaminated staple foods is widespread. Biomarkers such as aflatoxin-albumin complexes, urinary aflatoxin M_1_, and aflatoxin (AF) DNA adducts provide valuable insights but remain underutilized in resource-limited settings. Despite the globally recognized health risk posed by AF, research focused on monitoring human exposure remains limited, particularly among immunocompromised individuals. This dynamic emphasizes the need for targeted studies and interventions to address the particular risks faced by immunocompromised individuals. This review provides an up-to-date overview of AF exposure in immunocompromised populations, including individuals with cancer, hepatitis, diabetes, malnutrition, pregnant women, and the elderly. It also highlights exposure pathways, biomarkers, and biomonitoring strategies, while emphasizing the need for targeted interventions, advanced diagnostics, and policy frameworks to mitigate health risks in these vulnerable groups. Addressing these gaps is crucial to reducing the health burden and developing public health strategies in high-risk regions.

## 1. Introduction

Aflatoxins, toxic secondary metabolites produced predominantly by the genera *Aspergillus*, particularly *Aspergillus flavus* (*A. flavus*) and *Aspergillus parasiticus* (*A. parasiticus*), are among the most potent mycotoxins and have serious public health implications [1,2]. The most important types of aflatoxins (AFs) include aflatoxin B_1_ (AFB_1_), the most potent and well-documented carcinogen among them, as well as aflatoxin B_2_ (AFB_2_), aflatoxin G_1_ (AFG_1_), and aflatoxin G_2_ (AFG_2_). Among these, AFB_1_ and AFG_1_ are considered highly toxic and mutagenic due to their strong genotoxic and hepatocarcinogenic properties [3,4]. These toxins are formed under environmental conditions such as high temperatures (25–35 °C), high humidity (≥70%), and during pre-harvest or improper storage of crops [2]. Other factors that favor their formation in agricultural products are insect infestation [5], drought stress [6], and improper food and feed handling. The biosynthesis of AF by toxic fungal species, particularly *A. flavus* and *A. parasiticus*, is a complex process regulated by a specific gene cluster in the fungal genome. Their biosynthesis occurs via a polyketide pathway in which the precursor, acetyl-CoA, undergoes enzymatic transformations to produce intermediates such as norsolorinic acid, leading to AFB_1_ and other variants [7,8].

Staple foods such as maize [9,10], peanuts [9,11] rice [12,13], sorghum [11,12], wheat [13,14], pearl millet [15], and milk in the form of aflatoxin M_1_ [10,16] are frequently contaminated with AF, leading to serious food safety and health concerns worldwide. Exposure to AF occurs primarily through ingestion, although inhalation, dermal contact, and transplacental ingestion are also possible routes [17]. Cao et al. [18] reported that the toxicity of AF is primarily mediated by its metabolic activation in the liver, where cytochrome P450 enzymes convert AFB_1_ into a highly reactive epoxide metabolite (AFB_1_-8,9-epoxide). This metabolite attaches to deoxyribonucleic acid (DNA), and forms adducts that induce mutations and genomic instability, thus contributing to carcinogenesis [18]. The negative effects of AF range from immediate toxicity to long-term health problems, such as liver cancer, gallbladder cancer, hepatotoxicity, genotoxicity, immunosuppression, growth retardation in children, and in severe cases, death [3,8,19,20].

As confirmed by Costa et al. [20], AFB_1_, the most dangerous and widespread AF variant, has been classified as a Group 1 carcinogen by the International Agency for Research on Cancer (IARC). While AF exposure poses a health risk to the general population, immunocompromised individuals such as those living with HIV/AIDS, cancer, diabetes, chronic hepatitis, autoimmune disorders, or organ transplant recipients are particularly at risk due to their impaired immune function [21,22,23]. AF-induced immunosuppression can impair cytokine production and reduce the activity of macrophages and natural killer (NK) cells, key components of the innate immune system [3,24]. Additionally, Chu et al. [25] reported that AFB_1_ exposure significantly increases the risk of hepatocellular carcinoma and liver cirrhosis, especially in individuals chronically infected with the hepatitis B virus.

In regions with high HIV prevalence and poor food safety practices, the combination of HIV-related immunosuppression and AF exposure represents a dual health crisis. This includes increased susceptibility to infection and disease due to weakened immunity [26], and an increased risk of AF-related health problems such as liver damage, hepatocellular carcinoma, and other immunodeficiencies [21,27].The global burden of AF is disproportionately prevalent in low- and middle-income countries (LMICs), where regulatory frameworks for food safety are weak, and contamination of staple foods is widespread [28]. These regions also have a greater incidence of immunocompromising illnesses, such as HIV/AIDS, making the intersection of AF exposure and immunosuppression an urgent public health challenge [29].

The lack of robust surveillance systems, combined with limited access to safe food, exacerbates the risks for vulnerable populations in these areas. Efforts to detect and monitor AF exposure have primarily focused on the use of biomarkers such as AF-albumin adducts [28,30], AFM_1_ in urine [13] and AF DNA adducts [31,32]. While these methods provide valuable insights into the extent of exposure, their application in immunocompromised populations has not yet been sufficiently researched. More importantly, prevention strategies for AF contamination have focused on improving agricultural practices [33], biological control of AF-producing fungi [34,35], chemical detoxification of AF [35], and proper post-harvest storage to reduce the growth of AF-producing fungal species [34].

However, these measures have not been able to combat the increased risks to immunocompromised individuals. Despite the globally recognized health risk posed by AF, there has been limited research focused on monitoring AF exposure in humans, let alone in immunocompromised individuals [36]. This gap emphasizes the need for targeted studies and interventions to address the particular risks to immunocompromised individuals. Therefore, this review provides a detailed overview of AF exposure in immunocompromised populations, including people with cancer, hepatitis, diabetes, malnutrition, pregnant women, and the elderly. It also provides a thorough and detailed examination of the various pathways of AF exposure in these vulnerable populations, including ingestion, carriage from animal products, in utero and maternal transmission, and inhalation. The review also looks in detail at the commonly used matrices and biomarkers for AF detection and explores the impacts of AF exposure on diseases such as diabetes, hepatitis, HIV, and cancer. Particular emphasis is placed on the increased susceptibility of vulnerable populations, including the elderly, pregnant women, malnourished individuals, and patients with HIV, diabetes, hepatitis, or cancer, who are at increased risk of AF-induced toxicity due to their weakened immune systems. More importantly, the review highlights future perspectives and emphasizes the need for targeted interventions, advanced diagnostic tools, and policy frameworks to reduce the health impact of AFs in this high-risk group. By bridging the existing knowledge gaps, this study aims to inform research priorities and public health initiatives and ensure the development of tailored approaches to safeguard the health of immunocompromised individuals in the face of this persistent threat.

## 2. Biosynthetic Pathway of AFs in *Aspergillus* Species

The biosynthesis of AFs occurs via a complex polyketide pathway, involving a series of oxidative, reductive, cyclization, and methylation reactions. According to Liao et al. [37], Ma et al. [38], Okechukwu et al. [8], and Yang et al. [39], the pathway begins with the condensation of acetyl-CoA and several units of malonyl-CoA through the activity of a polyketide synthase (PKS) enzyme complex encoded by the *aflC* (also known as *pksA*) gene. This reaction leads to the formation of the first stable intermediate, norsolorinic acid (NOR), a red pigment and an important early marker in aflatoxin biosynthesis. Norsolorinic acid (NOR) is reduced by *aflD* (*nor-1*) to form averantin (AVN). Subsequent hydroxylation and oxidation reactions mediated by enzymes such as *aflG* (*avnA*) and *aflH* lead to the formation of averufin (AVF). AVF is then further modified via several intermediate steps involving key enzymes such as *aflI* (*ver-1*), *aflV* (*cypX*), and *aflW* (*moxY*) to form versiconal hemiacetal acetate (VHA), which cyclizes into versicolorin A (VERA). VERA is a crucial branching intermediate that ultimately leads to the formation of demethylsterigmatocystin (DMST) and subsequently to sterigmatocystin (ST) through oxidative reactions. ST, an intrinsically toxic compound, is then o-methylated by O-methyltransferase A (*aflP*, also known as *omtA*) to O-methylsterigmatocystin (OMST).

The final steps of the pathway involve a series of oxidation and rearrangement reactions catalyzed by enzymes such as *aflQ* (*ordA*) and *aflU* (*cypA*), which convert OMST into the four major AFs: AFB_1_ and AFG_1_, followed by the reduced forms AFB_2_ and AFG_2_. The structural differences between B- and G-type AFs are due to variations in the oxidation state and the lactone ring structure [4]. AFB_1_ is the most potent and widely studied aflatoxin, known for its strong hepatocarcinogenic properties in humans and animals. This biosynthetic pathway operates within a large gene cluster spanning approximately 70 kilobases and includes around 25–30 genes, many of which encode enzymes for each step of the pathway [40]. Although this section focuses on the biosynthetic sequence, it is important to recognize that the expression of these genes is highly dependent on environmental factors such as temperature, moisture content, substrate composition, and oxidative stress. Understanding the enzymatic steps and key intermediates in this pathway provides a basis for developing strategies to monitor, detect, and mitigate AF contamination in agricultural products.

## 3. AF Exposure Pathways in Humans

Despite extensive research and ongoing efforts to tighten food safety regulations, exposure to mycotoxins remains high worldwide. An estimated five billion people remain at risk, with populations in sub-Saharan Africa disproportionately affected [41]. This increased vulnerability is largely attributed to favorable climatic conditions for fungal growth, recurrent food insecurity, and poor implementation of legislation to control mycotoxin contamination in food, with some regions of Asia having been regularly exposed [42]. Although people can come into contact with mycotoxins through various pathways (see Figure 1), the global extent of this exposure is still not well understood. This data gap limits our ability to fully understand health implications. To date, more than 400 mycotoxins have been discovered; only a small number are considered a significant threat to public health and agriculture, particularly in sub-Saharan Africa [43,44]. The tropical environment of the region, with its harsh and erratic climate, combined with traditional farming methods, makes the crop highly susceptible to mold infestation. This, in turn, leads to the formation of toxic compounds such as AF [45]. Cereal grains are one of the main food categories that are particularly susceptible to AF, but dried fruit, nuts, coffee, and spices are also affected. Intake of AF-contaminated foods and/or animal-derived foods (such as milk and eggs), tissues, and dairy products is the main pathway of AF exposure, with a significant proportion of AF being absorbed dermally and through inhalation. Otherwise, direct exposure to AFs occurs via consumption of contaminated food/feed and indirect exposure via animal foods such as tissues, eggs, milk, and other dairy products from animals fed with AF-contaminated feed as well as through dermal contact and inhalation of mold-contaminated dust.

### 3.1. Exposure to AF Through Ingestion

Aflatoxin exposure represents a substantial public health challenge, particularly in regions with inadequate food safety controls and suboptimal storage conditions. Aflatoxin risk assessment is a structured and methodical approach aimed at evaluating the potential health hazards associated with exposure to this group of food contaminants. The assessment process generally comprises four key components: hazard identification, exposure assessment, dose-response assessment, and risk characterization [46]. According to Kibugu et al. [47], hazard identification involves determining the adverse health effects associated with AFs, particularly AFB_1_. Exposure assessment quantifies the levels and frequency of aflatoxin ingestion through diet across various populations. The dose-response assessment establishes the relationship between the magnitude of exposure and the likelihood of specific toxicological outcomes, such as hepatotoxicity and liver cancer [48]. Risk characterization integrates data from the previous steps to estimate the overall health risk, with particular attention to vulnerable populations, including young children, the elderly, and immunocompromised individuals.

Exposure to AF through ingestion remains a significant global food safety concern, particularly in areas where staple foods are susceptible to fungal contamination. In 2015, the World Health Organization (WHO) stated that approximately 600 million people become ill each year due to contaminated food, resulting in 420,000 deaths and the loss of 33 million healthy life years [49]. Among the greatest threats to food safety are mycotoxins, secondary metabolites produced by toxic fungi such as *Aspergillus*, *Penicillium*, and *Fusarium* species [50]. In response, various nations have established stringent regulations to limit mycotoxin exposure. The Joint FAO/WHO Expert Committee on Food Additives estimates the average dietary exposure to AFs at 0.93 to 2.45 ng/kg/day in Europe, 3.5 to 180 ng/kg/day in Africa, and 0.3 to 53 ng/kg/day in Asia [51]. Africa in particular has the highest incidence of liver cancer, which is largely due to chronic ingestion of AFs. Previous estimates suggest that around 250,000 deaths in China and Africa are due to hepatocellular carcinoma [52]. Dietary exposure to AFs is influenced by consumption habits, processing methods, and geographical factors. Staple foods such as maize, peanuts, and other grains are highly susceptible to AF exposure, especially in low-income regions. For instance, researchers have indicated extremely high rates of AF contamination in food (see Table 1). Alberts et al. [53] indicate that the daily maize intake in some rural communities reaches 1–2 kg per person, which significantly increases the AF load. AFs have also been detected in processed foods, including fermented beverages [54], and cheese [55]. Despite the widespread assumption that food processing eliminates mycotoxins, studies confirm that AFs exhibit high thermal and chemical stability, rendering conventional food processing methods insufficient for complete detoxification [56,57]. Consequently, contamination by AF in various foods remains an ongoing concern emphasizing the need for continuous monitoring and effective remediation strategies.

#### 3.1.1. Exposure Through Carryover from Animal Sources

Farmers often feed mold-contaminated and physically damaged crops to livestock, assuming they are unsuitable for human consumption but still usable as animal feed. However, these crops may contain high levels of mycotoxins, particularly AFs, which pose serious health risks to animals [74,75]. AFs. Various types of mycotoxins have been detected in animal feed, including AFs, which are usually associated with cereals and nuts that make up the majority of feed ingredients or formulations [76]. Depending on the animal class and/or species, there is a possibility that microorganisms in the rumen affect the concentration of AFs transferred to animal tissues, leading to consumer exposure through the consumption of tissues contaminated with AFs [77,78]. Monogastric species, for example, are more susceptible to the effects of AFs compared to ruminants.

This is because the microbiota in the rumen of ruminants degrades most AFs [79]. Nevertheless, the ability of the rumen to detoxify AFs may also be limited within species, age, as well as sex, resulting in some AFs accumulating in animal tissues and representing a potential source of exposure for humans if these contaminated tissues are consumed [80]. Therefore, the proportion of AFs in the consumable tissues of ruminants is comparatively low, as some of the AFs are degraded in the rumen, while others are processed in the liver, and therefore have no significant impact on human exposure compared to non-ruminants [80]. AFs ingested by animals can either be excreted in urine, excreted in milk, or assimilated into tissues. As mentioned above, AFs are low in edible tissues, and milk contaminated with mycotoxins has been identified as one of the main sources of human exposure [81,82].

However, the degree of carryover is also influenced by dietary factors such as feeding, intake rate, digestion rate, health status of the livestock, age, sex, ability to biotransform in the liver, husbandry systems, seasons, geographical location, and environmental conditions that affect the extent of AF carryover in animals. Consumption of eggs and meat from animals fed with AF-contaminated feed is another possibility of indirect exposure. Numerous research papers have documented the effects of AFs on fish [83,84,85], and AF-contaminated feed for fish in aquaculture has been reported [86]. When such AF-contaminated feed is consumed, humans and other animals are exposed to AFs.

#### 3.1.2. In Utero and Maternal Exposure

Research has shown that exposure to AFs in utero can occur via the transplacental route [38,87]. Although breast milk is rich in nutrients and immune factors, it can also be a potential source of exposure to AFs for infants. The hydroxylated derivative of AF (AFM_1_) is found in breast milk 12–24 h after consumption of AF-contaminated foods. Magoha et al. [88] analyzed the breast milk of 143 Tanzanian lactating mothers for AFM_1_ and reported 100% contamination. Similarly, Mahdavi et al. [89] reported AFM_1_ levels in the breast milk of Iranian breastfeeding mothers who had exclusively breastfed their children. Waseem et al. [51] found that 99.5% of the breast milk of 445 mothers from Abu Dhabi was contaminated with AFM_1_, and 36% of Egyptian mothers were contaminated with AFM_1_. High concentrations of ochratoxins and AFs were also identified in Italian mothers. Infant formula was a source of AF contamination.

Exposure in utero and through the mother is therefore a feasible source of human exposure to AFs. The first 1000 days spent alive, starting from conception to 24 months of age, are an important period for the growth and development of the child. Therefore, food intake during pregnancy plays a fundamental role in the future health of the child when it comes to the mother’s consumption of food contaminated with AFs [90,91]. In addition, children of weaning age are considered a high-risk group for AF exposure. Foods suitable for weaning include corn and peanuts which have been reported to be particularly susceptible to mycotoxin contamination (Table 1).

### 3.2. Exposure to AF Through Inhalation

Many studies have shown that exposure to AFs is not always related to food consumption and that the work and residential environments (see Table 2) are also potential sources of AF exposure exposure [92,93,94,95,96,97]. The nasal passage is the principal target for many inhaled toxins, with the epithelial mucosa being the first to be damaged by these inhaled spores or AFs [98]. Systemic effects of AF through inhalation of some AFs have been reported such as lung cancer, whose mechanism of carcinogenicity is due to oxidative DNA damage [95,98,99]. Although this is the least studied route of exposure, some reports have shown that breathing in a portion of AFs may pose greater risks than ingestion. Exposure to AFs in the environment or workplace is mainly through inhalation. Airborne dust often contains fungal spores and hyphae fragments and can transport AFs to the lungs via inhalation, which is common in the workplace [98,100]. Work environments with poor ventilation subject employees to an increased danger of aflatoxin contamination [93]. Environmental exposure to AF through molds in water-damaged buildings is well known. Various mycotoxins have been detected in human biopsies of lung, brain, and liver tissue and body fluids of people who have been exposed to toxin-producing molds in their environments [76].

### 3.3. Exposure to AF Through Dermal Contact

As diet is the primary factorial method regarding exposure to AF, exposure from the work environment or occupation should not be disregarded as in some cases it may also contribute directly to the overall exposure. When AF-containing dust particles come into contact with the skin, either through contaminated work surfaces or in situations where workers must frequently handle AF-contaminated materials such as food or feed without protection, exposure to AFs is likely [111].

Workplaces, especially in agriculture and the food industry, including tasks like loading, handling, or milling of AF-tainted substances (grain, waste, and feed), as well as feed processing plants, are good examples of aflatoxin exposure sites, as most aflatoxin concentrations are 10 times higher than in human food. Work environments with inappropriate protective clothing and equipment put workers at higher risk of AF contamination [81]. Taevernier et al. [112] have shown that mycotoxins can penetrate the skin and cause skin damage. Therefore, sporadic exposure to AF also increases the risk of cell death in the epidermis, skin cancer, and immune-related diseases [111].

## 4. AF Biomarkers

AFs are among the most commonly detected fungal toxins in food and animals, resulting in their status as the most biomonitored mycotoxins [113,114,115,116]. The metabolism of AFs, particularly AFB_1_, along with biochemical reactions such as hydroxylation and epoxidation, determines the resultant forms of these compounds. AFM_1_ is a thoroughly studied metabolite and a confirmed indicator of AFB_1_ exposure, detectable in various biological liquids such as urine and breast milk [117]. Wild and Gong [118] highlighted several biomarkers used to monitor AF exposure, including urinary metabolites such as AFM_1_, AFB_1_-N7-Gua, AFP_1_, AFQ_1_, and AFB_1_-mercapturic acid, as well as AF-albumin adducts [119].

Understanding the sources of AF exposure and their metabolic pathways is essential for the advancement of appropriate biomarkers for exposure estimation. Furthermore, the effective use of biomarkers necessitates the availability of detailed information regarding individual AF toxicokinetics, bioavailability, and metabolism to accurately interpret the results [120]. Human data on toxicokinetic studies for various toxins are limited, as most studies have predominantly been conducted in animal models; consequently, exposure assessment using biomarkers presents a quantitative approach fraught with significant uncertainties [121]. Biomonitoring includes the detection of AFs in readily accessible biological liquids like blood, milk, feces, and urine (see Figure 2). The main benefit of these biomarkers rests in their capacity to assess AF exposure levels within biological matrices, which integrate all previously identified sources of [122,123]. The detection of biomarkers can be conducted using one of the following approaches: direct biomarkers, which involve the application of particular standardized methods that are refined and confirmed methods utilizing parent compounds as reference substances [124], indirect biomarker quantification, which entails assessing non-specific structural or functional alterations in the body induced by exposure to certain drugs or toxins; or non-targeted biomarkers, which involve the measurement of unknown aflatoxin derivatives.

Biomarkers for AF exposure have been identified in various biological fluids (see Table 3) and tissues, including milk, plasma, saliva, feces, hair, nails, liver, kidneys, and lungs [125]. Among these biomarkers, urine is the most utilized matrix, not only because it contains a wide array of AFs but also due to the ease of obtaining urine samples through non-invasive methods [126]. Different biomarkers are suitable for varying purposes and contexts; for instance, breast milk is particularly effective for evaluating aflatoxin exposure in breastfed infants, whereas other biological fluids, such as serum and plasma, require more invasive collection methods and trained medical personnel, making them more applicable for long-term exposure studies [126]. Thus, the biomonitoring of AFs in mycotoxin exposure can be conducted across diverse vulnerable groups, thereby reducing assumptions about dietary exposure [127]. Determining the precise level of AFs within bodily fluids highlights the ambiguity associated with the presence of emerging or modified AF types that might go unnoticed by traditional analytical methods, particularly those mycotoxins lacking legal limits and/or defined biomarkers, which present additional challenges. The following section will outline commonly used biomarkers, along with their respective advantages and disadvantages.

### 4.1. Urine

AFs are excreted in urine as metabolites following the ingestion of contaminated food. After absorption in the gastrointestinal tract, AFs, particularly AFB_1_, are transported to the liver, where they undergo biotransformation primarily by cytochrome P450 enzymes [8,128]. This hepatic metabolism converts AFB_1_ into several hydroxylated metabolites, including aflatoxin M_1_ (AFM_1_), aflatoxin Q_1_ (AFQ_1_), and aflatoxin P_1_ (AFP_1_). These metabolites are subsequently released into the bloodstream, filtered by the kidneys, and excreted in urine as part of the body’s detoxification and elimination processes [129].

Urine provides a convenient, non-invasive, and efficient matrix for the biomonitoring of aflatoxin exposure in humans [130,131]. Following the ingestion of aflatoxin-contaminated food, the compounds undergo hepatic metabolism, resulting in the formation of metabolites such as AFM_1_, AFQ_1_, AFP_1_, and AFB_1_-N7-Gua, which are subsequently excreted in urine. Quantification of these metabolites allows for an accurate assessment of recent aflatoxin exposure. Due to the ease of urine collection, this approach is particularly suitable for large-scale epidemiological studies, including those involving high-risk groups such as children, the elderly, HIV-infected individuals, and malnourished populations [132,133]. To enhance the reliability of exposure estimates, urinary aflatoxin concentrations are commonly normalized to creatinine levels, thereby accounting for interindividual variability in physiological factors such as age, sex, body mass, and hydration status [134,135]. Collectively, urine-based biomonitoring offers a robust, accessible, and scientifically sound method for evaluating aflatoxin exposure across diverse demographic cohorts.

Hydroxylation at the C-7 position results in the production of AFM_1_, while epoxidation at the C-8 and C-9 positions results in either internal or external configurations. The exo-epoxidized type of AFB_1_ can react with DNA and glutathione or undergo hydrolysis to form AFB-diol [96]. Specifically, AFB_1_-N7-guanine and AFB_1_-glutathione conjugates are excreted in urine as AFB_1_-mercapturic acid conjugates, serving as indicators of exposure [136,137]. Panel et al. [138] also reported measurable levels of AFB_1_-N7-guanine in urine samples. Additionally, other metabolites such as AFM_1_, AFQ_1_, and AFBN7-Gua are excreted in both feces and urine as a result of AFB_1_ biotransformation mediated by specific cytochrome P450 isoforms. To assess exposure, urinary AF concentrations are converted to levels of intake that account for their kinetics, body weight, and the daily output of urine, primarily characterized by creatinine levels [134,136,137].

Variability in urine volume among individuals can influence the levels of AFs and their metabolites that are excreted, with daily fluctuations in aflatoxin intake posing a significant challenge [134]. This issue can be mitigated by normalizing aflatoxin levels to creatinine concentrations, as the AF/creatinine ratio allows for comparisons between individuals, accounting for factors such as muscle mass, sex, age, and season [139]. Thus, urinary creatinine concentrations serve to adjust the urinary concentrations of AFs or their metabolites. Furthermore, the analysis of AFs and their metabolites in urine samples requires low LOQs, which can be achieved through targeted analysis and/or enzymatic treatment of urine samples with β-glucuronidase/sulfatase to convert conjugates back to the parent compound [115].

### 4.2. Blood

Blood-based biomonitoring is a critical tool for evaluating long-term aflatoxin exposure. Repeated ingestion of AFs leads to the formation of aflatoxin–albumin adducts, which circulate in the bloodstream for extended periods due to the stability and longevity of albumin [3,140]. Quantifying these adducts in serum or plasma provides a reliable indicator of chronic exposure. This technique is particularly beneficial for assessing exposure in high-risk populations, such as individuals with cancer, diabetes, or HIV, who may exhibit increased susceptibility to the adverse effects of AFs [138]. While conventional blood sampling is invasive, the use of dried blood spot (DBS) methods offers a minimally invasive and more practical alternative, especially in remote or resource-limited settings where access to clinical infrastructure is restricted. DBS facilitates effective population-level monitoring without requiring extensive medical equipment or personnel.

AFs are produced by fungi like *Aspergillus flavus* in contaminated food. After ingestion, AFs are absorbed in the gut and transported to the liver. In the liver, they are metabolized by cytochrome P450 enzymes into various toxic and non-toxic forms. Some metabolites, such as AFM_1_, AFQ_1_, AFP_1_, and AFB_1_-8,9-epoxide, enter the bloodstream. In blood, AFs can bind to albumin, forming aflatoxin–albumin adducts used as biomarkers. These circulating metabolites are later excreted in urine or bile.

AF-albumin adducts have long been used as a substitute efficacy biomarker of AFB_1_ exposure for assessment due to the long half-life of albumin in humans [141,142]. Reports indicate that over 95% of blood samples in various regions of West Africa contain AF-albumin adducts as biomarkers, encompassing individuals of all ages. The advantages of these biomarkers include their higher levels of toxins, making them particularly useful in long-term exposure studies [126,143]. While serum and plasma are not commonly employed as routine biomarkers due to the invasive techniques required for collection, a non-invasive method for detecting AFB_1_-albumin adducts from dried blood spots has been reported [144].

### 4.3. Breast Milk

Breast milk plays a vital role in assessing recent exposure to AFs, especially AFM_1_, which appears following maternal ingestion of AFB_1_-contaminated food. This is particularly concerning for breastfeeding infants, whose immature detoxification systems make them highly vulnerable to toxin exposure [145]. Measuring aflatoxin levels in breast milk enables accurate evaluation of exposure in both lactating mothers and their infants, groups often overlooked in traditional dietary assessments [146]. Consequently, breast milk monitoring provides critical insights into aflatoxin-related health risks for neonates and young infants, supporting targeted interventions to protect these sensitive populations.

Lactating mothers and their infants constitute a particularly vulnerable population due to the potential transfer of AFs from the maternal system to breast milk. When a lactating woman consumes aflatoxin-contaminated food, mainly AFB_1_, it is absorbed and metabolized in the liver. The liver converts AFB_1_ into AFM_1_, a hydroxylated metabolite. AFM_1_ is then transported through the bloodstream to the mammary glands. It is excreted into breast milk, especially within 12–24 h of exposure [90,147]. AFM_1_ in breast milk is heat-stable and toxic, though less potent than AFB_1_. The presence of AFM_1_ in breast milk provides a direct route of exposure for breastfeeding infants, who are especially susceptible due to their developing organs and immature detoxification systems. Monitoring AFs in breast milk is critical for assessing exposure levels in lactating mothers and their infants, guiding public health interventions, and mitigating associated risks [148,149]. Detecting and quantifying AFs in breast milk necessitates sensitive and reliable analytical techniques.

Breast milk serves as a valuable matrix for estimating the exposure of breastfed infants to AFs; however, its limitation lies in its applicability solely to lactating women. AFM_1_ is the predominant biomarker of exposure, having been detected in both milk and urine during exposure assessments [150]. However, while breast milk analysis provides critical data on infant exposure, its applicability is limited to lactating women, and the concentration of AFM_1_ in milk is influenced by maternal dietary habits, metabolic efficiency, and interindividual differences in detoxification pathways [80,140]. Detecting and quantifying AFM_1_ in breast milk requires highly sensitive and reliable analytical techniques such as HPLC, ELISA, and LC-MS/MS. These methods ensure accurate assessment of exposure levels, contributing to risk mitigation strategies aimed at protecting maternal and infant health.

### 4.4. Feces

AFs are excreted in feces as unmetabolized toxins and metabolites. After ingestion, AFs are absorbed and metabolized mainly in the liver. Some unmetabolized AFs and their metabolites are secreted into the bile which transports these compounds into the intestines [151]. From the intestines, AFs and metabolites are eliminated via feces.

Fecal biomarkers enable the assessment of recent aflatoxin exposure by detecting unmetabolized toxins and their metabolites excreted via the gastrointestinal tract. This non-invasive approach is practical for large-scale population studies; however, its validity and applicability in vulnerable populations, such as the elderly, pregnant women, and children, remain to be fully established. The analyses and effective method for detecting AFs and their metabolites in humans and animals provide valuable insights into short-term dietary exposure and metabolism [152]. AFB_1_, the most toxic and prevalent aflatoxin, is often detected in feces in its unmetabolized form, serving as a direct indicator of recent dietary intake [147,151]. In addition to AFB_1_, feces contain a range of metabolites, including AFQ_1_ and AFM_1_, which result from hepatic biotransformation processes. Notably, higher levels of AFQ1 are often detected in feces compared to AFM_1_, indicating a significant excretory route for metabolism [147,151]. Since feces reflect unabsorbed AFs and metabolic byproducts excreted through bile, fecal biomarkers provide a complementary assessment of systemic exposure and detoxification efficiency [152].

The presence of AFM_1_ is particularly relevant for populations consuming AF-contaminated food, as it highlights both dietary intake and metabolic processing capacity [153,154]. The detection of these adducts is invaluable for assessing genotoxicity, as persistent exposure to AFs has been linked to hepatocellular carcinoma (HCC) and other adverse health effects [138,155]. Using fecal biomarkers is especially advantageous for large-scale biomonitoring studies, as sample collection is non-invasive, and fecal matter provides a stable medium for mycotoxin analysis. Recent studies have demonstrated the potential of fecal biomarkers to reflect both direct dietary intake and metabolic processing of AFs.

For instance, a 2024 study investigated the effects of AFB_1_ on human intestinal Caco-2 cells, identifying critical genes affected by AFB_1_ exposure [156]. The study highlighted changes in gene expression related to cell proliferation, migration, apoptosis, and metabolism, suggesting that these genetic alterations could serve as biomarkers for AF contamination detection. Integrating fecal biomarker assessments with other biological matrices, such as urine and blood, enhances the accuracy of exposure assessments and improves the understanding of interindividual variability in AF metabolism and excretion. As analytical techniques such as LC-MS and HPLC continue to advance, fecal biomarker analysis will remain a critical tool in mycotoxin research and public health monitoring.

**Table 3 toxins-17-00414-t003:** Matrices and AFs biomarkers commonly used in exposure assessment.

Matrix	Biomarker	Technique	Sample Size	% Contamination	Concentration Range (ppb)	References
Urine	AFM_1_	HPLC	72	29.17	0.00067–0.00787	[157]
Urine	AFM_1_	HPLC	69	78.26	0.0006–0.0399	[158]
Urine	AFM_1_	LC-MS/MS	220	14	0.06–4.7	[159]
Feces	AFB_1_	PLE and HPLC-MS/MS	3	1	0.02	[160]
Urine	AFM_1_	HPLC	300	27.67	0.01–0.33	[161]
Urine	AFM_1_	LC-ESI-MS/MS	145	10.34	0.17–1.38	[162]
Urine	AFB_1_-N7-Gua	HPLC-MS/MS	20	80	0.9–7.2	[163]
Urine	AFB_1_-N7-Gua	HPLC	27	40.74	6.6–494.9	[164]
Urine	AFM_1_	ELISA	205	57.56	0.00020–0.0193 creatinine	[165]
Urine	AFs	2-D TLC	60	100	0.07–0.2	[166]
Urine	AFM_1_	HPLC	50	64	0.008–0.801	[167]
Urine	AFM_1_	ELISA	93	47.31	500–59,900 creatinine	[168]
Urine	AFM_1_	ELISA	160	61.25	LOD-0.0747	[169]
Urine	AFM_1_	HPLC-ESI-MS/MS	175	-	0.005-0.5	[170]
Urine	AFM_1_	HPLC-MS/MS	28	10.74	LOD-0.33	[171]
Urine	AFM_1_	LC-MS/MS	120	14.2	0.3–1.5	[172]
Plasma	AFB_1_-lys	LC-MS/MS	260	19.6	10.5–74.5	[115]
Plasma	AFB_1_ AFB_2_	LC-MS/MS	60	39.4 74.91	1.23–4.56 1.16–3.75	[173]
Plasma	AFB_1_-lys	LC-Orbitrap	58	23.45	0.2–59.2	[174]
Serum	AFB_1_-lys	LC-FLDLC-MS/MS	34	83	1.08–102.6	[31]
Plasma	AFB_1_-lys	LC-MS	32	46.88	-	[175]
Serum	AFB_1_-lys	LC-MS	160	61	0.80–20.24	[176]
Serum	AFB_1_-lys	LC-FLD	220	100	0.71–95.6	[177]
Serum	AFB_1_-lys	LC-FLD	347	99.4	-	[178]
Serum	AFB_1_-lys	LC-FLD	884	100	6.04–8.90	[179]
Serum	AFB_1_-lys	LC-MS	461	100	0.2–814.8	[180]
Plasma	AFB_1_-lys	LC-MS/MS	60	72	3.5–6.6	[181]
Plasma	AFB_1_-lys	LC-MS/MS	85	-	-	[181]
Plasma	AFB_1_-lys	LC-MS/MS	167	62	0.04–123.5	[182]
Plasma	AFB_1_-lys	ELISA	115	100	3.9–458.4	[183]
Serum	AFB_1_-lys	ELISA	230	67	-	[184]
Serum	AFB_1_-lys	ELISA	305	88.2	-	[185]
Plasma	AFB_1_-lys	ELISA	374	95	-	[186]
Serum	AFB_1_ AFM_1_	LC-MS/MS	213	23 50	0.0–0.73 0.0–1.91	[187]
Serum	AFB_1_-lys	LC-FLD	713	90	0.4–168	[188]
Plasma	AFB_1_-lys	ELISA	166	67–98	4.7–23.50	[189]
Serum	AFB_1_-alb	ELISA	34	98	3.0–35.1	[168]
Blood/serum	AFB_1_-alb	ELISA	24	100	LOD-32.8	[190]
Blood/serum	AFs	2-D TLC	60	100	0.15–0.38	[166]
Blood/serum	AFB_1_-alb	HPLC	507	100	0.44–268.73	[191]
Blood/serum	AFB_1_-alb	ELISA	250	98	LOD-66	[192]
Blood/serum	AFB_1_-alb	LC-MS/MS	597	78	0.02–211	[193]
Blood/serum	AFB_1_-alb	HPLC	170	97	0.2–23.16	[194]
Blood/serum	AFM_1_	HPLC	131	39	0.3–56	[195]
Blood/serum	AFB_1_-alb	ELISA	119	100	4.8–260.8	[196]
Blood/serum	AFB_1_-alb	HPLC	170	20.6	1.01–16.57	[197]
Breastmilk	AFM_1_	HPLC	388	36	5.5–5131	[198]
Breastmilk	AFM_1_	HPLC	75	100	60.9–299.9	[199]
Breastmilk	AFB_1_	HPLC	75	100	94.5–4123.8	[199]
Breastmilk	AFM_1_	HPLC	140	92	5–3400	[200]

## 5. AF Biomonitoring in Immunocompromised Individuals

Biomonitoring of AFs in at-risk populations involves the detection of AF metabolites in biological samples such as urine, serum, and breast milk [201]. Techniques such as LC-MS, HPLC, and ELISA have been widely used for this purpose [202]. The diagram (Figure 3) illustrates key factors contributing to the susceptibility of immunocompromised individuals to AF toxicity. It categorizes vulnerable groups, including pregnant women, HIV patients, diabetes patients, cancer patients, and elderly individuals, highlighting their specific risk factors that compromise detoxification and immune responses to AF exposure. Therefore, regular biomonitoring can provide insights into the exposure levels of immunocompromised individuals helping in formulating dietary interventions to reduce risks [203,204].

### 5.1. Elderly Individuals

The natural decline in immune function with age is one of the main contributors to the vulnerability of elderly individuals to the adverse effects of mycotoxins. The role of an intact immune system and barrier function in the successful elimination of pathogenic fungi and their toxic metabolites cannot be overemphasized [205]. Mycotoxins are implicated in facilitating a decline in the body’s natural defenses. For example, the *A. flavus* toxins AFB_1_ and AFM_1_ compromise the intestinal epithelial cell barrier by inhibiting intestinal cell growth and viability. Since these epithelial cells act as the body’s primary defense against fungal toxins, their disruption facilitates inappropriate immune responses such as chronic inflammation of vital organs [206]. Additionally, disturbances in the enzyme and antimicrobial agent secretory functions of epithelial cells and poor nutrient retention have been reported by Gonkowski et al. [207].

This double burden, namely the natural decline in immune function and the additional compromise caused by mycotoxin exposure, places the elderly at a heightened risk of developing severe health complications. Biomonitoring in this population becomes critical as an opportunity to identify early signs of exposure and implement timely interventions. Strategies such as routine screening for mycotoxin biomarkers and dietary adjustments to reduce exposure can help mitigate health risks [141]. However, a critical limitation of existing studies is the exclusion of elderly individuals over 60 years of age. Available studies tend to group all adults into a single category, that is, those aged 18 years or older. This limits insights into mycotoxin exposure in the elderly immunocompromised population, which is the focus of this review.

In a mycotoxin exposure biomonitoring study by Patriarca et al. [201] using the urinary multi-biomarker approach, different analytes were detected using LC–ESI–MS/MS in 110 out of 175 samples collected from Cameroonian adults (up to 58 years of age) in Bamenda and Yaoundé. Among the mycotoxins identified, AFM1 was a key compound detected, contributing significantly to the overall percentage of analytes found in urine samples. Notably, an HIV-positive adult in the study was found to be exposed to a mixture of multiple mycotoxins, including AFM_1_, highlighting the relevance of this particular mycotoxin in the exposure assessment. An assessment of the food consumption practices of the same study population revealed the consumption of maize and cassava-based products such as corn *fufu*, *garri*, rice, plantain, cocoyam, pumpkin, and potatoes. These dishes are generally accompanied by groundnut, *egusi*, or vegetable soup. Typically, combined maize and groundnut-based foods were consumed once to five times a week in both Bamenda and Yaoundé. Maize and groundnuts are mycotoxin-vulnerable foods, which explains the high mycotoxin exposure rates in the elderly in these two regions of Cameroon [201]. Based on the roughly calculated amounts of mycotoxins ingested per day from these foods, the urinary levels of AFM_1_ were estimated to exceed the recommended tolerable daily intake (TDI) limits proposed by the 2003 and 2013 Scientific Committee on Food.

In another study by Huang et al. [198], urine specimens from 227 adults in China aged 20–88 years were analyzed for mycotoxin exposure. The biomarker for AFB_1_ and AFM_1_ was noted at a low prevalence in only 2.20% of the specimens (mean ± SD: 0.35 ± 0.17 ppb). Despite the low occurrence, given that AFM_1_ is a class 1 carcinogen and highly toxic even at lower concentrations, it is still necessary to monitor and reduce these levels to non-detectable concentrations. While young adults (20–45 years), middle-aged individuals (46–64 years), and senior adults (65–88 years) are all prone to mycotoxin contamination, this same study showed that mycotoxins were more frequently detected in young adults, possibly due to a greater food intake compared to older adults. The relatively low prevalence and concentration of toxins in samples from young adults could be due to a high detoxification rate of toxins, which limits their bioavailability [208]. This may suggest a decline in detoxification capacity and efficiency with age. Interestingly, females were found to be more prone to AFM_1_ exposure than males. Correlation studies between urinary AFM_1_ concentrations and food consumption indicated that exposure was linked to the consumption of aflatoxin-prone foods, such as seeds and nuts.

### 5.2. Pregnant Women

Breast milk serves as a valuable biological matrix for assessing the vulnerability of infants to mycotoxin contamination. However, urine and plasma samples are also effective for monitoring mycotoxin exposure during pregnancy. Exposure to multiple mycotoxins in early pregnancy can pose significant health and developmental risks to the fetus, as these toxins can cross the placental barrier [209]. For instance, in utero exposure has been linked to low birth weight and, in some cases, stunted growth. Specifically, mycotoxin exposure is associated with differential gene methylation, affecting growth and immune-related genes, therefore potentially impairing the child’s development and immune function [210].

A 2018–2019 study of 447 pregnant women in rural Bangladesh revealed exposure to AFs, among other mycotoxins of interest in the study. While AFs were not among the most prevalent in the study population, only 17 of the samples were clear of mycotoxin exposure. These figures indicate widespread exposure of rural populations in Bangladesh to mycotoxins, which, if unmonitored, can have adverse effects on pregnancy outcomes. Market studies in Bangladesh have revealed high consumption of mycotoxin-prone foods such as rice, wheat flour, groundnuts, spices, dates, and milk, with mycotoxin levels in some exceeding U.S. maximum regulatory thresholds [211]. Comprehensive mycotoxin control strategies must be prioritized across the entire value chain to mitigate exposure risks effectively. This includes promoting effective agricultural practices, proper post-harvest management, storage, and transportation to inhibit fungal growth and mycotoxin production contamination. Additionally, enforcing country-specific regulatory limits, conducting regular surveillance, and raising public awareness of mycotoxins and their impacts are crucial to ensuring food safety.

Simultaneous detection of mycotoxins using LC-MS/MS in biological fluids, serum (*n* = 71), urine (*n* = 18), and amniotic fluid (*n* = 21) samples collected from pregnant women in Alessandria, Italy, demonstrated that mycotoxins could be present in fetal-maternal biological fluids. However, the detected levels were too low to pose significant risks to the mother or fetus. Specifically, AFG_1_ was detected in one serum and four urine samples and AFB_1_ and AFB_2_ were detected in two urine samples [212]. In contrast, an assessment of biological samples from 98 pregnant women in Egypt revealed a mean serum AF-albumin (AF-Alb) contamination of 4.9 pg/mg albumin in 34 of the samples and a urinary AFM_1_ mean concentration of 19.7 pg/mg creatinine in 44 of 93 samples. Essentially, AFs were detected in 41% of the subjects studied. In another study, an ELISA analysis of 80 human breast milk samples obtained from clinics in Jordan revealed AFM_1_ contamination in most samples, with only three falling below the maximum tolerance standards established by the European Union and U.S. regulations [213]. These findings highlight the variation in AF exposure across different regions, emphasizing the need for country-specific regulations and tailored control strategies to effectively address localized risks. Also, the prevalence of AFs in breast milk poses a major risk for the conveyance of toxins to infants, potentially leading to serious health issues.

Southern African countries are among the regions most at risk for mycotoxin contamination due to the frequent consumption of cereal-based foods and groundnuts. However, research on exposure assessment and biomonitoring of AFs among pregnant women in this area appears to be limited. Smith et al. [180] conducted a study to determine aflatoxin exposure among pregnant women in Zimbabwe by measuring urinary AFM_1_ levels. Urine specimens were gathered from 1580 participants in rural regions, specifically the Chirumanzu and Shurugwi regions in the Midlands Province, and analyzed for AFM_1_ using direct ELISA. The study found that 484 women, or 30%, were exposed to aflatoxin, with AFM_1_ mean values ranging from 31 to 6046 pg/mg. The greatest levels of exposure were identified in areas with lower elevations and limited rainfall. The consumption of high-risk items like maize, peanuts, and milk was linked to the observed mycotoxin exposure rates in this study [180].

### 5.3. Malnourished Individuals

Malnutrition, particularly in children, remains a critical health burden in developing economies. The widespread prevalence of AF exposure in these regions further exacerbates the problem, as these toxins impair nutrient absorption, compromise immune function, and contribute to stunted growth and developmental delays [206,214]. A study conducted in Nigeria involving 58 children (aged 6 to 48 months) from both rural and urban communities revealed alarming findings [174]. Among these children, 47 were diagnosed with severe acute malnutrition or undernutrition (SAM), with 21 suffering from marasmus and 26 suffering from kwashiorkor: additionally, 43 exhibited stunting. Notably, 81% of these children had measurable levels of AFB_1_-Lys adducts in their serum, with concentrations ranging from 0.2 to 59.2 pg/mg albumin. Children with SAM and stunting had significantly higher concentrations compared to the control group without malnutrition. Furthermore, AFB_1_-Lys adduct levels varied notably between children with kwashiorkor and those with marasmus, with kwashiorkor patients exhibiting the highest concentrations. Although AF exposure may not directly cause kwashiorkor or marasmus, evidence suggests that malnutrition significantly impairs the body’s ability to detoxify mycotoxins [181]. This is supported by the lower mycotoxin concentrations observed in non-malnourished children, as reported by [174].

On the contrary, Gong et al. [215] reported a connection between chronic AF exposure and growth impairment and underweight in children from Benin and Togo. Interestingly, AF levels were low in breastfed infants, suggesting that children are most vulnerable to mycotoxin contamination post-weaning, primarily through contaminated solid foods. This highlights an urgent need for integrated strategies to address both malnutrition and mycotoxin exposure in children to improve health outcomes, especially in vulnerable populations.

### 5.4. AF Biomonitoring in HIV Individuals

Biomonitoring of AF exposure in HIV-infected individuals is essential for assessing potential health risks and understanding its implications for disease progression. Biomarkers such as AF-albumin adducts provide valuable insights into long-term exposure, particularly in resource-limited settings where dietary contamination with AFs is common [216,217]. The connection between AF exposure and HIV progression has been increasingly studied, with evidence suggesting that chronic exposure may exacerbate immune suppression, increase viral replication, and contribute to associated comorbidities.

Recent studies highlight the high prevalence of AF exposure among HIV-infected populations, particularly in sub-Saharan Africa. For instance, a cross-sectional study conducted in Nigeria reported that 95.9% of 196 HIV-positive patients had detectable levels of urinary AFM_1_, indicating widespread exposure [218]. Similarly, research by Jolly et al. [219] demonstrated that high baseline AFB_1_ albumin adduct levels were significantly linked to lower CD4 T-cell counts in ART-naïve HIV patients, with an average count of 66.5 cells/µL and a statistically significant p-value of 0.043. These findings suggest that AF exposure may accelerate HIV-related immunosuppression, contributing to increased susceptibility to opportunistic infections, liver dysfunction, and complications associated with disease progression.

The interaction between AFs and HIV infection is mediated through multiple mechanisms. AFs have well-documented immunosuppressive properties that impair immune cell function, further weakening the already compromised immune system of HIV-positive individuals [220]. The compromised immune response may lead to faster progression of HIV-related complications. This suppression can reduce the body’s ability to fight opportunistic infections, leading to more severe disease outcomes. Additionally, some studies suggest that AF exposure is associated with increased HIV viral replication, potentially due to the disruption of cellular pathways involved in viral regulation [221]. Furthermore, AFs induce DNA damage through the formation of reactive metabolites that bind to DNA, leading to mutations. In the context of HIV, this mutagenic potential may influence viral evolution and contribute to the development of drug resistance, complicating antiretroviral therapy (ART) efficacy. AF exposure is also known to contribute to malnutrition, which can further impair immune function and reduce ART effectiveness, making disease management more challenging. The health implications of aflatoxin exposure in HIV patients are significant. Immunosuppression caused by AFs increases the risk of opportunistic infections, leading to higher hospitalization rates and greater healthcare burdens. Additionally, the interaction between AFs and HIV may complicate treatment strategies, as patients with higher aflatoxin exposure could experience reduced ART effectiveness, necessitating individualized treatment modifications. Chronic exposure to AFs also raises concerns regarding long-term health outcomes, particularly the heightened risk of hepatocellular carcinoma, a concern for HIV-positive individuals who are already at an elevated risk for liver disease.

Given these risks, routine biomonitoring of AF exposure in HIV patients, especially in endemic regions, is critical. Implementing public health strategies such as improved food storage practices, dietary diversification, and targeted interventions to reduce aflatoxin contamination could help mitigate these risks [222]. Further research is needed to establish definitive causal relationships and explore intervention strategies aimed at minimizing aflatoxin exposure in HIV-vulnerable populations.

### 5.5. AF Biomonitoring in Diabetes Patients

Diabetes mellitus, characterized by chronic hyperglycemia, is associated with immune dysfunction and an increased susceptibility to infections. The immunocompromised state of diabetic individuals makes them particularly vulnerable to environmental and dietary toxins such as AF, that have been implicated in a variety of negative health impacts, such as immunosuppression, hepatotoxicity, and carcinogenesis [223,224]. AFB_1_ is recognized as a potent hepatocarcinogen and metabolic disruptor, and its effects in diabetic populations warrant careful investigation. Given that diabetes is already linked to a higher risk of metabolic disorders and infection-related complications, AF exposure may further amplify these health risks, creating a cycle of worsening disease outcomes [223,225]. The interaction between AF and diabetes involves multiple biochemical pathways that can exacerbate disease progression. One of the primary concerns is the impact of AFs on glycemic control. AF exposure has been connected to disruptions in glucose metabolism and insulin signaling, leading to insulin resistance and increased difficulty in maintaining blood sugar levels [226,227]. This disruption can exacerbate hyperglycemia in diabetic patients, complicating their condition [223].

Additionally, AFs induce oxidative stress, which damages pancreatic β-cells responsible for insulin production, thereby impairing insulin secretion and contributing to hyperglycemia [23,228]. Long-term exposure to AFs has also been shown to elicit inflammatory responses, further exacerbating insulin resistance and metabolic dysfunction [229]. The inflammatory cytokines produced in reaction to aflatoxin exposure can aggravate systemic inflammation, which is already a key factor in diabetes pathogenesis, leading to further metabolic imbalances.

In addition to metabolic disturbances, AF exposure may contribute to an increased risk of infections in diabetic patients. Given that diabetes itself impairs immune function, the immunosuppressive effects of AFs may further compromise the body’s capacity to combat infections, raising the likelihood of complications such as diabetic ketoacidosis and poor wound healing [225].

Moreover, long-term exposure to AFs has been linked to an elevated risk of cardiovascular disease, neuropathy, and retinopathy—complications that are already prevalent in diabetic populations [230,231]. Another critical concern is the potential interaction between AFs and antidiabetic medications. AF exposure may interfere with the metabolism and bioavailability of these drugs, reducing their efficacy and necessitating adjustments in treatment regimens [223]. Additionally, diabetic individuals are already at a higher likelihood of developing certain cancers, and long-term AF exposure further amplifies this risk, particularly for hepatocellular carcinoma [232].

The intersection of diabetes and AF exposure poses a considerable public health issue, especially in areas with high dietary AF contamination. Tackling this problem necessitates a multifaceted approach that includes routine biomonitoring of AF exposure in diabetic individuals, particularly those in high-risk regions. Public health initiatives should focus on food safety measures, improved agricultural practices, and dietary diversification to reduce AF exposure. Increased awareness among healthcare providers and patients is also essential for promoting preventive strategies and early intervention. Educational campaigns targeting both the medical community and the public can help improve understanding of the risks associated with AF exposure and encourage proactive measures to mitigate its impact [232]. While existing evidence highlights the adverse effects of AFs in diabetic individuals, further research is needed to fully elucidate the molecular mechanisms underlying these interactions and to develop targeted strategies for reducing exposure and associated health risks.

### 5.6. AF Biomonitoring in Cancer Patients

Cancer is defined as a complex medical condition characterized by the uncontrolled growth of cells, leading to tumor formation and, in many cases, metastasis to other organs [233]. The progression of cancer is influenced by various genetic, environmental, and lifestyle factors, including exposure to dietary carcinogens such as AFs. AFs, particularly AFB_1_, are potent hepatocarcinogens that undergo metabolic activation in the liver, leading to the formation of reactive epoxides that bind to DNA and induce mutagenic changes [2,8,234]. These interactions contribute to cellular damage, inflammation, and oxidative stress, all of which play a role in tumor initiation and progression. Cancer patients, particularly those undergoing chemotherapy or radiotherapy, are immunocompromised and may be more susceptible to the harmful effects of AF exposure due to their reduced ability to detoxify and repair cellular damage. The carcinogenic effects of AFs are mediated through their metabolic activation by cytochrome P450 enzymes in the liver, which convert this aflatoxin (AFB_1_) to highly reactive intermediate forms [2,8].

These metabolites interact with DNA, RNA, and proteins, leading to the creation of DNA adducts that trigger mutations and genomic instability, raising the chances of oncogenesis [235]. AF-induced DNA damage can further contribute to the disruption of critical cellular processes, including apoptosis and cell cycle regulation, thereby promoting the survival of malignant cells. Moreover, AFs induce oxidative stress through the excessive production of reactive oxygen species (ROS) that harm cellular macromolecules, disrupt metabolic pathways, and promote chronic inflammation [236,237]. This inflammatory environment facilitates tumor progression by enhancing cellular proliferation and angiogenesis, further supporting cancer development.

The immunosuppressive effects of AFs compound the challenges faced by cancer patients, as these toxins impair immune surveillance and reduce the body’s ability to eliminate pre-cancerous and cancerous cells [3,238]. Cancer patients undergoing chemotherapy or radiotherapy are particularly vulnerable, as these treatments already diminish the immune system’s effectiveness, making them more vulnerable to infections and other environmental toxins. Additionally, AFs can interfere with the metabolism of chemotherapeutic drugs, potentially reducing their efficacy and complicating treatment outcomes [239]. The interaction between AFs and other carcinogenic substances, such as alcohol and certain dietary components, may also contribute to an increased cancer risk by enhancing mutagenic potential and promoting tumorigenesis [240,241,242].

Furthermore, nutritional deficiencies common in cancer patients, including deficiencies in vitamins A, C, and E, may impair the detoxification of AFs, exacerbating oxidative stress and DNA damage [240,241,242].

The consequences for public health of AF exposure in cancer patients are significant, particularly in regions where food contamination is prevalent. Awareness and education about the risks associated with AF-contaminated food are crucial for minimizing exposure, particularly for immunocompromised individuals [3,239]. Strengthening food safety regulations and monitoring AF levels in food supplies are essential measures to protect at-risk populations, including cancer patients [243]. Further studies are necessary to better understand the effect of AF exposure on cancer progression and treatment efficacy, as well as to develop targeted strategies for reducing exposure and mitigating associated health risks. Implementing biomonitoring programs that assess AF levels in cancer patients could provide valuable insights into the relationship between dietary aflatoxin intake and disease outcomes, ultimately contributing to improved clinical management and public health interventions.

### 5.7. AF Biomonitoring in Hepatitis Patients

AF biomonitoring involves the detection and quantification of AF metabolites in biological samples like blood, urine, and tissues [244]. Hepatitis B or C patients are at heightened risk due to their weakened immune systems [245]. AFB_1_, the most potent hepatotoxic variant, has been linked to liver disease progression, leading to increased morbidity and mortality among these populations [3]. The hepatotoxic effects of AFs are primarily mediated through their metabolic activation in the liver. Upon ingestion, AFB_1_ undergoes bioactivation through the cytochrome P450 enzyme, leading to the development of highly reactive epoxide metabolites that form adducts with DNA and proteins [3]. This DNA damage triggers mutagenic processes, increasing the risk of liver cancer [2,8]. Additionally, AFs contribute to immune suppression, further impairing the body’s capacity to fight infections and exacerbating disease progression in hepatitis patients.

Various analytical techniques are employed for biomonitoring, including enzyme-linked immunosorbent assays (ELISA), high-performance liquid chromatography (HPLC), and mass spectrometry (MS) techniques [246]. These methods facilitate accurate exposure assessment, allowing for early detection and intervention in high-risk individuals (Table 4). Recent studies indicate a significant correlation between AF exposure and liver disease progression among hepatitis patients [247]). Research conducted in West Africa demonstrated that chronic hepatitis patients had significantly elevated levels of AF metabolites compared to healthy controls, emphasizing the importance of focused monitoring strategies in these vulnerable populations [248]. The integration of biomonitoring into routine clinical assessments could improve disease management and mitigate AF-associated health risks in immunocompromised individuals. Effective monitoring programs can identify high-risk populations and inform targeted interventions to minimize exposure to AFs [249,250]. Furthermore, strategies such as improved food storage practices, stricter regulatory frameworks, and enhanced surveillance systems can mitigate AF contamination.

Finally, raising awareness among healthcare professionals and at-risk patients, while strengthening collaborations between public health authorities, researchers, and clinicians, is essential for developing effective policies to mitigate AF exposure and enhance health outcomes in hepatitis patients.

## 6. Conclusions

AF biomonitoring in immunocompromised and high-risk populations is critical for assessing exposure risks and associated health outcomes. Elderly individuals, pregnant women, malnourished individuals, and patients with HIV, diabetes, cancer, or hepatitis are highly susceptible to aflatoxin-induced toxicity due to compromised immune function and metabolic vulnerabilities. AFs, particularly AFB_1_, are potent hepatotoxins and carcinogens, exacerbating liver damage in hepatitis and cancer patients while increasing the risk of hepatocellular carcinoma. In malnourished individuals, aflatoxin exposure impairs nutrient absorption and immune response, further worsening health outcomes. Advanced detection techniques, including LC-MS, ELISA, and HPLC, have improved AF identification in biological matrices, yet challenges remain in defining exposure thresholds and understanding chronic low-dose effects.

## 7. Future Perspectives and Recommendations

There is a wide range of immunocompromised individuals, but this review focuses on the most commonly affected groups: pregnant women, the elderly, cancer patients, individuals with diabetes, and those living with HIV. To enhance future research, additional vulnerable populations such as organ transplant recipients, patients with chronic kidney or liver disease, malnourished individuals, and young children should be considered to better address the broader challenges of immunosuppression. Addressing aflatoxin exposure in immunocompromised patients will require integrated strategies aimed at preventing further exposure, enhancing detoxification, and supporting immune function. Future research should prioritize real-time, non-invasive detection methods using sensors and wearable technologies for continuous biomonitoring. Nanotechnology is revolutionizing the detection of AFs through the development of highly sensitive methods. Nanosensors, utilizing materials such as gold nanoparticles and carbon nanotubes, enhance the detection of AFs in food and biological samples. These sensors offer rapid and efficient results, making them ideal for on-site testing, which is particularly beneficial for farmers and food safety inspectors. CRISPR technology is being repurposed for aflatoxin detection. This cutting-edge method exploits CRISPR’s precision to identify specific DNA sequences associated with aflatoxin-producing fungi. As a result, contamination can be detected early and accurately, thereby facilitating more effective food safety management.

Significant advancements are being made in the development of portable devices (Portable Analytical Devices) designed for on-site AF testing. These devices typically employ microfluidics and lab-on-a-chip technology, enabling efficient analysis without the complexity of traditional laboratory setups. This innovation is particularly advantageous for farmers and food safety inspectors in rural settings, allowing for immediate food safety assessments. However, artificial intelligence and machine learning are increasingly being applied to predict aflatoxin contamination in crops. By analyzing environmental variables and historical data, these technologies contribute to risk assessment and management. This proactive strategy enables farmers to implement preventive measures before contamination occurs, thus protecting both their crops and public health. Innovative biosensors are being developed that integrate biological recognition elements with electronic signal transducers to detect AFs, even at low concentrations. These sophisticated sensors can be tailored to target specific AFs and facilitate real-time monitoring of food safety, providing reassurance to both consumers and producers. Research is ongoing into novel detoxification methods, such as the use of biochar and other adsorbents, to address AF contamination in agricultural products. These techniques have the potential to significantly reduce aflatoxin levels in contaminated foods, enhancing their safety for consumption. Additionally, researchers are investigating genetic engineering as a strategy to develop crop varieties that are resistant to AFs. By modifying the genes that make crops vulnerable to aflatoxin-producing fungi, scientists aim to minimize contamination at its source, thereby promoting safer food production.

Omics-based approaches like metabolomics and proteomics can help elucidate biochemical changes caused by AFs, while longitudinal studies are needed to establish dose-response relationships and refine regulatory limits. Understanding the mechanisms of aflatoxin-induced immunosuppression is essential, especially with emerging AF-producing fungi such as *A. nomius* and *A. pseudotamarii,* which call for epidemiological studies and control measures. Climate change is expected to worsen aflatoxin contamination, emphasizing the need for climate-resilient agricultural practices and predictive models. Innovative solutions using nanotechnology for detection and prevention, along with advances in personalized medicine, can further enhance mitigation strategies. The development of rapid point-of-care diagnostic tests for aflatoxin biomarkers will facilitate timely interventions, while detoxification methods and nutritional support may reduce toxicity in vulnerable groups. Additionally, future efforts should focus on developing effective chemoprotective agents and antioxidants to reduce liver damage and oxidative stress. Early detection and monitoring of liver health and cancer risk will be crucial, as will tailored nutritional support to strengthen immunity and optimized management of underlying conditions like HIV. Improving patient education on food safety and implementing routine aflatoxin screening in high-risk populations will also be key to reducing exposure. Advances in these areas will help protect vulnerable groups from the harmful effects of AFs.

## Figures and Tables

**Figure 1 toxins-17-00414-f001:**
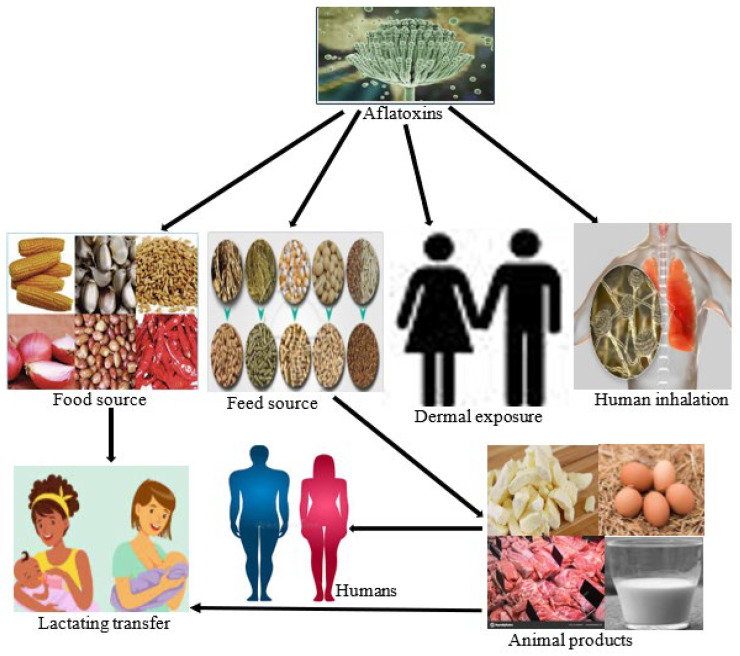
Various AF exposure routes in humans.

**Figure 2 toxins-17-00414-f002:**
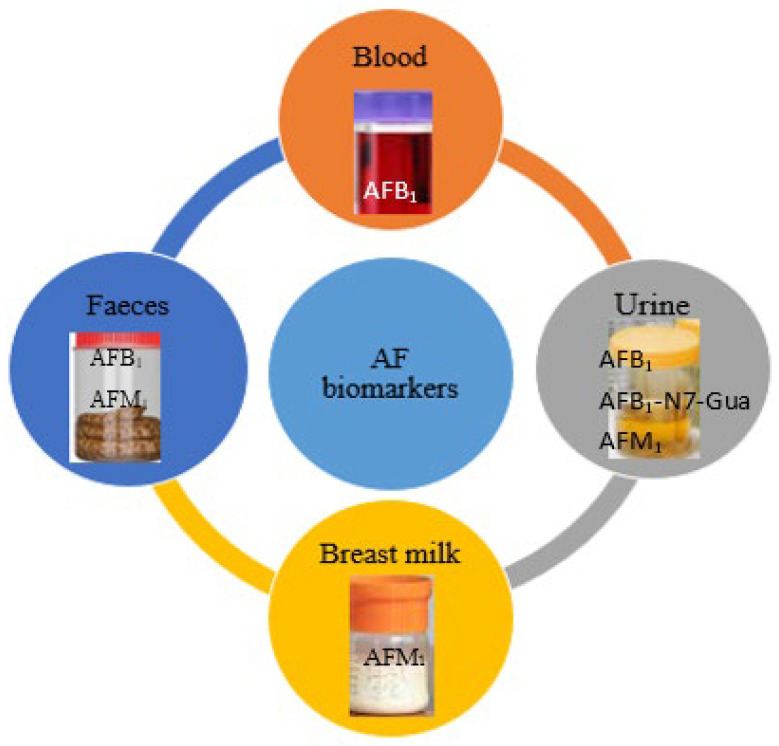
Biomarkers for AFs in biological samples.

**Figure 3 toxins-17-00414-f003:**
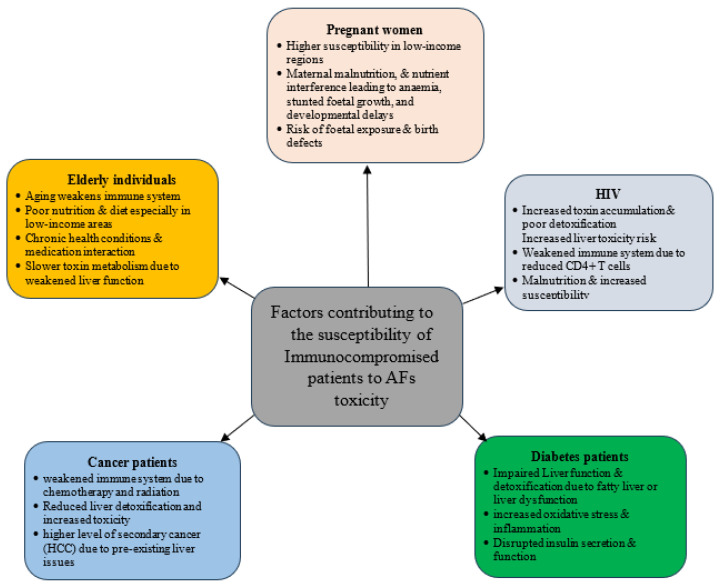
AF biomonitoring in immunocompromised individuals.

**Table 1 toxins-17-00414-t001:** Dietary exposure to aflatoxin.

Food Commodity	Type of Aflatoxin	Analytical Method	Sample Size	% Contamination	Concentration Range (ppb)	References
Maize	AFs	HPLC	200	80	LOD-40.31	[27]
Groundnut	AFs	HPLC	180	100	LOD-162.40	[27]
Breakfast cereal	AFB_1_	HPLC-FD	26	69	LOD-0.13	[58]
maize	AFs	HPLC	800	52	LOD-1369	[59]
Maize fermented dough	AFs	HPLC	32	97	LOD-310	[60]
Maize	AFs	HPLC	-	100	20–355	[60]
Maize	AFB_1_	LC-MS/MS	114	47	1–149	[61]
Ginger	AFs	HPLC, ELISA	100	100	3.63–411.1	[62]
Processed maize	AFB_1_	HPLC	176	50	-	[63]
Home-brewed beer	AFs	HPLC/TLC	29	28	200–400	[64]
Groundnut	AFs	ELISA	75	93	LOD-43.23	[65]
Dried beef	AFB_1_	LC-MS/MS	108	66	3.91–295.41	[66]
UHT Milk	AFM_1_	HPLC	11	54.5	0.013–0.219	[67]
Dairy products	AFM_1_	HPLC	156	45.5	0.015–7.350	[67]
Rice and maize	AFs	ELISA	32	74	1.75–173.3	[68]
Cowpea	AFs	LC-MS/MS	81	-	LOD-209	[69]
Sorghum	AFs	ELISA	20	100	4.80–42.60	[70]
Millet	AFs	ELISA	20	100	4.80–45.60	[70]
Yam flour	AFs	ELISA	20	100	5.0–39.45	[70]
Garri	AFs	ELISA	20	100	2.60–55.40	[70]
Rice	AFs	ELISA	62	100	2.10–248.20	[70]
Milk	AFM_1_	HPLC-FD	372	56.1	LOD-345.8	[71]
Maize	AFs	HPLC-FD	180	57	1.3–91.4	[72]
Maize	AFs		3000	5–72	LOD-76.2	[73]
Milk and dairy products	AFM_1_	ELISA	160	100	0.137–0.319	[55]

%—Percentage; AFs—total AFs; AFB_1_—aflatoxin B_1_; AFM_1_—aflatoxin M1; HPLC—High-Pressure Liquid Chromatography; HPLC-FD—High-Performance Liquid Chromatography with Fluorescence Detection; LC-MS/MS—Liquid Chromatography–Tandem Mass Spectrometry; ELISA—Enzyme-Linked Immunosorbent Assay; TLC—Thin-Layer Chromatography; LOD—Limit of Detection.

**Table 2 toxins-17-00414-t002:** Environmental and occupational exposure to AF.

Environment	Analytical Method	Sample Size	% Contamination	Contamination Range (ppb)	References
Sugar production factory	ELISA	15	-	6–11	[101]
Sugar and paper-making factory	ELISA	181	56	5.9–10.4	[102]
Warehouses for green coffee, black pepper, and cocoa beans	HPLC	44	-	LOD-0.023	[103]
Farms handling grains	HPLC	24	-		[104]
Animal feed production	ELISA	45	20	LOD-8	[105]
Waste industry	ELISA	41	100	2.5–25.9	[106]
Poultry production	ELISA	31	59	1–4.23	[107]
Textile industry		58	33		[108]
Swine production	HPLC-MS/MS	25	16		[92]
Waste management	ELISA	41	-	-	[109]
Feed mill workers	HPLC	28	100	73.4–189.2	[110]

**Table 4 toxins-17-00414-t004:** AF biomonitoring and its effects on diabetes, hepatitis, HIV, and cancer.

Condition	Population	Biomarkers Assessed	Type of AF	AF Levels ppb	Method Used	References
Diabetes	15–55	Glucose levels, Glycosylated, Heamoglobin (HbA1c)	AFM_1_	1.86	Elisa kits, Microplate Elisa readers, Spectrophotometric methods	[23]
Pregnant women						
Hepatitis	Children, age 5–12	Liver enzymes (ALT, AST)	AFM_1_	15	Serum Biochemical Tests, Enzyme Immunoassays, Liver Function Tests	[251]
HIV	Adults with HIV-positive	CD4 count	AFB_1_	10	Flow Cytometry, Immunoassays, Laboratory Blood Tests	[252]
Cancer	Cancer patients, various types	Tumor markers	AFB_1_	25	Serum Biochemical Tests, ELISA	[253]
HIV	HIV	HIV Viral Load	AFB_1_	400 ng/kg body weight/day	Serum Biochemical Tests	[220]
Cancer	Cancer patients in European population		AFB_1_	10 ng/kg body weight per day	Margin of Exposure (MOE) approach	
Cancer	Hepatocellular Carcinoma in sub-Saharan Africa	HCC	AFB_1_	5 to 500	Quantitative cancer risk assessment	[254]
Cancer	Hepatocellular carcinoma in Southern Africa	HCC	AFB_1_	Peanut butter (6.8–250.) Peanut (6.6–622.1)	(HPLC)	[255]

## Data Availability

No new data were created or analyzed in this study. Data sharing is not applicable to this article.

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
