# Peer review of "Aflatoxin Exposure in Immunocompromised Patients: Current State and Future Perspectives"

_toxins, 2025, doi:10.3390/toxins17080414_

Round 1

Reviewer 1 Report

Comments and Suggestions for Authors

This paper reviews the current status of aflatoxin exposure, health risks, and monitoring strategies in immunocompromised patients, including the biological characteristics of aflatoxins, their formation environment, various exposure pathways, and biomarker detection technologies. It also elaborates on the specific risks faced by different immunocompromised populations. Through case analyses, the paper examines the current situation and hazards of aflatoxins and identifies key directions for future research. However, issues such as partial content repetition and inconsistent table formatting exist.

Specific comments:  

-The introduction's description of AFs' carcinogenic mechanisms (e.g., the cytochrome P450 metabolic pathway) could be supplemented with more recent studies to avoid over-reliance on pre-2020 data.

- In Table 1, the "Concentration range" lacks unit annotation—is it consistent with Tables 2 and 3?  

- The "% contamination" in Tables 1 and 2 differs in wording from "Rate of contamination (%)" in Table 3.  

- There are two instances of "Table 3" in the text—is this a labeling error?  

- In "3. AF biomarkers," Sections 3.1–3.3 discuss the rationale and applicability of urine, blood, and breast milk as biomarkers, whereas Section 3.4 on feces only mentions their advantages as markers without specifying the applicable population.  

- Is the classification of immunocompromised populations comprehensive enough?  

- The description of emerging technologies is relatively brief.  

- The terms "aflatoxin B1" and "AFB1" are used interchangeably, with multiple subscript errors (in tables and text).  

- The references are outdated, with limited inclusion of recent studies.

Author Response

Comment 1 : The introduction description of AFs carcinogenic mechanisms (e.g., the
cytochrome P450 metabolic pathway) could be supplemented with more recent studies to avoid over-reliance on pre-2020 data.

Response 1: We appreciate the reviewer comments. A recent reference has been added to AFs  carcinogenic mechanisms in the entire manuscript.

Comment 2: In Table 1, the "Concentration range" lacks unit annotation—is it consistent with Tables
2 and 3?

Response 2: We value the reviewer’s observation. The concentration unit of the Afs in tables 1 to 3 is
now consistence (ppb).

Comment 3: The % contamination; in Tables 1 and 2 differs in wording from Rate of
contamination (%) in Table 3.

Response 3:We value your through observation and review of our manuscript. The rate of contamination
(%) in table 3 has now be changed to “% contamination” to avoid confusion and maintain consistency.

Comment 4:There are two instances of Table 3 in the text—is this a labeling error?

Response 4: Thank you for this important observation. One of the two “table 3” has now been changed to
“table 4”.

Comment 5: In ''3 AF biomarkers, Sections 3.1–3.3 discuss the rationale and applicability of urine, blood, and breast milk as biomarkers, whereas Section 3.4 on feces only mentions their advantages as markers without specifying the applicable population.

Response 5: Thank you for your feedback on Section 3 of the manuscript. I appreciate your observation
regarding the discussion of feces in Section 3.4. I have clarifed the applicable population for fecal biomarkers in the revised version. Your insights are invaluable, and I am grateful for your thorough review.

Comment 6: Is the classification of immunocompromised populations comprehensive enough?

Response 6: Thank you for your comments. There is a wide range of immunocompromised individuals, but
this review focused on the most commonly affected groups: pregnant women, the elderly, cancer patients, individuals with diabetes, and those living with HIV. We have now recommended in lines (851 – 856) that future research should consider additional vulnerable groups such as organ transplant recipients, patients with chronic kidney or liver disease,
malnourished individuals, and young children to better address the broader challenges of immunosuppression.

Comment 7: The description of emerging technologies i relatively brief.

Response 7: Thank you for your valuable feedback. We have expanded the section on emerging technologies
to provide a more comprehensive overview. Specifically, we have included detailed
descriptions of recent advancements, their mechanisms of action, and their relevance to the
study's context

Comment 8: The terms "aflatoxin B1" and "AFB1" are used interchangeably, with multiple subscript errors (in tables and text).

Response 8:We have now corrected these mistakes in the entire manuscript. 

Comment 9: The references are outdated, with limited inclusion of recent studies

Response 9: Many thanks for your observation, this has been addressed.

Reviewer 2 Report

Comments and Suggestions for Authors

1. How did the author demonstrate the  biomarkers for AFs in biological samples? There are only images of biological samples in the figure 2 and the legend says that biomarkers. Authors can put specific biomarkers along with the sample bottles in the same figure.

2. How are AFs biosynthesized in these samples?

3. What metabolic pathways are being used to biosynthesize AFs.

4. What could be the treatment approach of Aflatoxin Exposure in Immunocompromised Patients?

Author Response

Comment 1: How did the author demonstrate the biomarkers for AFs in biological samples? There are only images of biological samples in the figure 2 and the legend says that biomarkers. Authors can put specific biomarkers along with the sample bottles in the same figure.

Response 1: 

We appricate your advice and your comment, this has been addressed on figure 2

Comment 2: How are AFs biosynthesized in these samples?

Response 2: Aflatoxins are produced by fungi in contaminated food. After ingestion, they are metabolized in the liver into various forms like AFM₁. These metabolites are then excreted into feces, blood, milk, and urine, reflecting exposure rather than biosynthesis. The metabolism and excretion of aflatoxins in each of these samples have now been included. 

Comment 3: What metabolic pathways are being used to biosynthesize AFs

Response 3:The metabolic pathways of Afs biosynthesise has now been included in the manuscript (lines 122 – 151). 

Comment 4: What could be the treatment approach of Aflatoxin Exposure in Immunocompromised Patients?

Response 4: 

We sincerely appreciate your comments. We have now recommended some of the treatment approach to minimize aflatoxin exposure in immunocompromised patients in lines 856 – 878.

Reviewer 3 Report

Comments and Suggestions for Authors

The paper describes the current state of aflatoxin exposure in immunocompromised patients, e.g. persons with HIV/AIDS. Aflatoxins are one of the potent mycotoxins that  pose important public health risks. The authors present an up-to-date review of the dietary exposure, the environmental and occupational exposure, the ways of biomonitoring aflatoxins using biomarkers. They also present results in more detail for five risk groups, pregnant women, malnourished individuals, persons with HIV, persons with diabetes, and persons with cancer. The finish the paper with some conclusions and future perspectives.

The authors did a great job in collecting information on the exposure and biomonitoring  of aflatoxins, both in general, and for the five risk groups in more detail. However, the paper needs an extensive check on the way is has been written down. In terms of sections (contents, titles, ordering), but mainly in terms of English grammar. See below.

I

Comments on the Quality of English Language

6: ‘one the’: ‘of' missing
9; ‘temp’?
14: ‘those’: ‘to those’
21: which gaps?
32: ‘avenues’-> ‘ways’
36: ‘including’-> ‘resulting in’
38: ‘the most potent carcinogen’: ‘most potent carcinogen of all aflatoxins’
43: ‘insect damage’: ?
45: ‘AF biosynthesis’: what is exactly meant here: the formation, de-formation, or both?
68-76: re-formulate these lines, too much repetition
82-91: what adds this section to the previous one?
99: ‘explore’: ‘explored’
108: ‘dynamic underscores’:?
121:122: how do I have to interpret these lines?
121:130: re-formulate this section
133: one ‘)’ too many
Section 2.1: it is about exposure, but starts with health risks
172:173: re-formulate line
Table 1: some comments: unit definition for the concentration range, describe all abbreviations fully, what is difference between Afs, and total Afs?
196: ’age and sex’?
219: ’12-24’:?
238: ‘are potential’: ‘are also potential’
Table 2: first row: .5ppb-.027ng/m3?
Section 3 on AF biomarkers. I would start with describing the metabolism process (described in section 3.1), resulting in biomarkers that can be measured, and then go into more detail on these biomarkers, instead of the other way around.
Section 4.3: in the title the term ‘malnourished’ is used, in the figure ‘elderly’; that is confusing
Sections 4.1-4.7: sometimes the term ‘AF biomonitoring is used, sometimes not; I would restrict to the population sub-group names, to stay in line with the figure
Section 4.7: not included in the figure

Reviewer 4 Report

Comments and Suggestions for Authors

A very interesting and complex study that follows the pathways of aflatoxin exposure in humans, by ingestion, carry-over from animal sources, in utero and maternal exposure, by inhalation. Very well done and complex tables with dietary exposure to aflatoxin including LOD values, , environmental and occupational exposure to Aflatoxin, Matrix and Aflatoxin biomarkers exposure commonly used and Aflatoxin biomonitoring and its effects on diabetes, hepatitis, HIV, and cancer. The study highlights the increased vulnerability of elderly people, pregnant  women, malnourished people, and patients with HIV, diabetes, cancer or hepatitis which are highly susceptible to aflatoxin-induced toxicity due to compromised immune  function. A special mention should be added to the very complex and topical bibliographical study.

Author Response

Comment 1: A very interesting and complex study that follows the pathways of aflatoxin exposure in humans, by ingestion, carry-over from animal sources, in utero and maternal exposure, by inhalation. Very well done and complex tables with dietary exposure to aflatoxin including LOD values, environmental and occupational exposure to Aflatoxin, Matrix and Aflatoxin biomarkers exposure commonly used and Aflatoxin biomonitoring and its effects on diabetes, hepatitis, HIV, and cancer. The study highlights the increased vulnerability of elderly people, pregnant women, malnourished people, and patients with HIV, diabetes, cancer or hepatitis which are highly susceptible to aflatoxin-induced toxicity due to compromised immune function. A special mention should be added to the very complex and topical bibliographical study

Response 1 : We sincerely thank the reviewer for their thoughtful feedback and for recognizing the complexity and value of our work. In response to the suggestion, we have incorporated the recommended information in lines 110–120.

Round 2

Reviewer 2 Report

Comments and Suggestions for Authors

Authors have revised the manuscript appropriately.